# Thinking together: How group argumentation boosts fake news recognition

**Diana Carbone** *, **Francesco Marcatto, Donatella Ferrante**

Department of Life Sciences, University of Trieste, Trieste, Italy

* diana.carbone@units.it

## Abstract

In a study with 111 college students, we present preliminary evidence that group discussion may improve the ability to recognize fake news. In the study, participants judged the veracity of news stories before and after participating in a group discussion or writing an individual reflection. The results showed that accuracy improved only in the case of fake news discussed in groups, while individual reflection was not sufficient to improve performance. An improvement in accuracy for individually evaluated fake news was observed only when these news items were presented after the group-discussed fake news in the experimental order, consistent with a transfer effect of group discussion on accuracy. Group argumentation produced positive effects on accuracy even in group interactions with inaccurate participants. The findings are in line with the Argumentative Theory of Reasoning.

## Introduction

On Aug. 25, 1835, the New York Sun published a shocking news story: the moon is inhabited by strange bat-men, winged unicorns and other fantastic creatures. The article cited the Edinburgh Journal of Science as the source and recounted the phantom discovery by a famous astrologer of the time, John Herschel. In fact, Richard Locke, the author of the article, had satirical intent but recounted the findings so rigorously that this news was picked up by the international press. This episode went down in history as "The Great Moon Hoax" and shows how misinformation is not a new phenomenon [1]. Nevertheless, social media has contributed to an exceptional spread of fake news, and the negative consequences it has on society and public health have given it unprecedented importance [2].

### The misinformation scenario

Terms such as fake news and misinformation, introduced in the previous paragraph, are now widely used in everyday discourse and are often employed interchangeably. However, both academic and political debates have highlighted the conceptual ambiguity surrounding these terms, and a shared definition has yet to be reached [3].

**Data availability statement:** All data, study materials, and analysis scripts underlying the findings of this study are publicly available on the Open Science Framework (OSF): https://osf.io/sgz2f/overview?view_only=45b-2ca98790043ed90b6cf7d590deb6e).

**Funding:** The author(s) received no specific funding for this work.

**Competing interests:** The authors have declared that no competing interests exist.

The term fake news is a broad and non-technical label that can encompass different types of information, ranging from unintentional errors and satirical content to deliberate and malicious manipulation and fabrication [4]. In the present study, this term is used as a generic descriptor for news-like content containing false information, without reference to the intentionality of its source. Consistently, we have used the term misinformation referring to false or inaccurate information shared without the deliberate intention to deceive [5,6] as the intent behind the false information used as study material is neither assessed nor manipulated.

Decades of research on human cognition have shown that misinformation is not easy to correct. According to a recent meta-analysis of 32 separate studies, the average effect of false information on beliefs even after exposure to corrections remained, albeit weak, present and significant [7]. In other words, it is often ineffective to be passively informed that something is false [8–11]. According to dual-process theories of thinking, the correction of misguided lower-level intuitive processes requires the mobilization of System 2. Bago and colleagues [12] provided evidence that if a quick assessment of a headline was followed by an opportunity to rethink, trust in fake – but not true – news was reduced. Pennycook & Rand [13] found a positive correlation between higher scores on the Cognitive Reflection Test [14] and the ability to discern between fake and real news in the context of accuracy judgments. Bronstein et al. [15] showed that dogmatic individuals, and religious fundamentalists were more likely to believe fake news (but not real), and that these relationships may be fully explained by analytic cognitive style. Moreover people who engaged in cognitive reflection were more discerning in their social media use in general, for instance by sharing higher quality content from more reputable sources [16]. However, there is also a long tradition in the study of analytical thinking that has shown that cognitive effort is not always beneficial per se: Even when relevant knowledge is available and the defense of one's own point of view is not a priority for the individual, analytical thinking and reflection may not be sufficient to ensure an improvement in the quality of judgements. People fail to generate instances that contrast the hypothesis they are explicitly testing, even when they know it is false, and sometimes they can fail to recognize flagrant counterexamples [17,18]. Kuhn and Modrek [19] showed that when people are asked to select evidence that contradicts a causal claim, they choose a counterclaim without evidence or evidence of an alternative sufficient cause, rather than the evidence of a failure of the alleged cause to produce the outcome. According to these authors, two main types of error affect our argumentation skills: The tendency to use single cause mental model of causality and to believe that an explanation of how the hypothesized cause could produce the outcome is valid evidence to support the hypothesized cause. Trouche and colleagues [20] showed that only 5% of participants changed their mind after verbally justifying their answers, and that participants who gave the intuitive but wrong answer were highly confident not only in their answer but also in the faulty reasons for the answer. However, when participants had the opportunity to discuss the previously individually considered reasoning problems in small groups, it was more likely that their reasoning errors were corrected.

### The social dimension of reasoning

Ascribing a predominant social function to reasoning is an idea whose origins reach far back into the past, into philosophy with Socrates and into psychology with Vygotsky and the Soviet socio-cultural school. More recently, developmental and educational psychologists demonstrated in multiple studies that dialogic argumentation is a successful means for the development of both individual and dialogic argumentative skills [21]. According to the Argumentative theory [22,23] reasoning has evolved to produce and evaluate arguments, with group discussions providing the ideal context for reasoning to perform effectively. Several studies have demonstrated that group discussions improve performance in different types of tasks, such as lie detection [24], forecasting economic and political events [25], solving biological problems [26], making medical diagnoses [27] and making judicial decisions [28].

### The current study

The aim of this study is to test whether, in a collaborative environment where the common goal of group discussion is accuracy, the exchange of arguments strengthens analytical thinking and improves the ability to recognize fake news. Specifically, we tested the effect of the group argumentation by having the participants judge the truthfulness of some news items before and after they had taken part in a group discussion or written their own argumentation. Our main prediction was that participants' ability to recognize fake news would improve after discussing the news in the group settings. As a second aim, in line with Trouche and colleagues [20], who have shown that people are sometimes able to transfer the understanding gained in a group discussion to other reasoning tasks with the same structure, we tested whether group discussions positively influence the subsequent individual ability to recognize new fake news.

## Materials and methods

### Participants

An a priori power analysis was conducted using G*Power 3.1 [29] for the primary hypothesis testing a 2 (news type: real vs. fake) × 2 (phase: 1 vs. 2) repeated-measures ANOVA design. We assumed a small-to-medium effect size of $f = 0.25$, based on effect sizes reported in the literature on fake news interventions (e.g., [30], with an alpha level of .05 and desired power $(1 - β)$ of .80. The analysis indicated a required total sample size of 128 participants to detect the hypothesized interaction effect.

Accordingly, 128 participants were recruited among college students and divided into groups of three. Seven participants did not show up on the scheduled date for the experiment, leading to the exclusion of their groups (14 participants in total). Additionally, one group of three was excluded due to poor internet connection affecting group discussion quality. Thus, the final sample consisted of 111 participants (mean age = 21.2 years, range = 18–32, SD = 2.6), with 79% female and 21% male. Participants received college credit as compensation for their participation.

This study received ethical approval from the University Ethics Committee (minutes n. 116, dd. October 4, 2021).

### Stimuli

Participants were asked to read and judge two real news and two fake news items. To select the news items, we reviewed articles from print publications, online newspapers, conspiracy websites, and debunking websites. As COVID-19 was a relevant topic at the time of data collection (between October 10, 2021, and March 23, 2022), we decided to select two news items about it and two news items on other topics. In general, the news items were selected to ensure diversity in topic (e.g., health, curiosities), temporal relevance (i.e., both recent and less recent news), and salience (Covid-related versus non-Covid news). Furthermore, all items were required to contain debatable content; news items that merely reported factual events were excluded, as such items offer little opportunity for discussion or reflection. For example, a straightforward report on a crime is difficult to argue as true or false. In contrast, items addressing broader topics rather

than isolated events allow for multiple perspectives and encourage deliberation and debate. The first fake news, "Form-aldehyde", came from a conspiracy website and claimed that the Covid vaccine contains formaldehyde, which has been linked to tumor manifestations. The second fake news, "Thyroid," claimed that mammograms increase the risk of thyroid cancer. The first real news, "Nutella," revealed that a Canadian company makes a special version of Nutella containing marijuana. Lastly, the second real news, "Masks," was an article from an Italian newspaper highlighting that the repeated use of surgical masks causes more harm than not using them at all. To test whether the selected messages were clearly written, appeared interesting and did not lead to strong judgements about truth/falsehood, we conducted a pre-test on the four messages. Twenty participants were asked to judge the truthfulness of each message, to rate on a 7-point Likert scale the confidence in their judgement of the truthfulness, the clarity of each message, and their interest in hearing others' views on the topic. The results of the pre-test showed that the accuracy of the truth judgements for the four messages was between 30% and 70%, the confidence average scores were between 3 and 5, the average scores for the clarity of the messages were above 5 and the average scores for the interest in hearing other people's views on the topic were between 3 and 5. The full texts of the four news items and the detailed results of the pre-test can be found in the supplementary material (S1 File and S1 Table).

## Procedure

The study was conducted online using a Google Forms online survey and Microsoft Teams. Participants were divided into groups of three and each group was given a date and time to connect via Microsoft Teams. The study adopted a within-subjects design and included two main phases: phase 1 and phase 2.

**Phase 1.** After receiving general instructions, participants were first asked to provide a written informed consent to participate and then individually complete an online questionnaire in which the four news items were presented. The participants had to read the items one after the other and answer the corresponding questions. The first two questions related to the intention to share the news (*"If you happened to see this news on social media, would you share it?")* and the judgement of truthfulness (*"Do you think this news is true or false?"*). The answers were given on a 6-point scale ranging from 1 (*most likely no / most likely false*) to 6 (*most likely yes /most likely true*). These two questions were always asked in this order to avoid a priming effect of truthfulness on sharing intention. Then, three additional questions were asked to control for social variables (*"Do you think this news might be of interest to your friends?"*), emotional aspects (*"Do you think this news has an emotional impact on readers?"*) and behavioral components (*"If you had the opportunity, would you look for more information online about this news?"*). The answers to these three questions were also given on a 6-point scale.

The four news items were presented in a partially counterbalanced design (i.e., Latin square) in which real news alternated with fake news and each news item was presented only once in each position.

**Phase 2.** Participants received a link to a new questionnaire in which the news items were displayed again. After each item, detailed instructions were given on what participants should do with each news item. In particular, they were asked to reread each news item and argue their viewpoint on the truth of the news item either in writing (individual task) or in interaction with other members of their group (group interaction). The instruction for the individual argumentation was: *"What aspects of this news indicate that it is true? What aspects suggest that it is fake?* State your reasoning with a brief argument."* It was an open-ended question where participants could write as much as they wanted. The instructions for the group's argumentation were: "What aspects of this news suggest that it is true? Which that it is fake? Discuss your point of view on this news item with other group members." The participants had 6 minutes for the group argumentation. They began by presenting their arguments one after the other in the order determined by the experimenter and could then use the remaining time to continue the discussion. This structured opening phase ensured that every participant had the opportunity to express their view, reducing potential effects of dominance, interruptions, or social loafing. The experimenter was visible only during the initial instruction phase and kept the camera off during the discussion; they

intervened only when strictly necessary (e.g., in the event of prolonged silence). All members contributed at least once during the initial turn-taking phase. Almost all groups used the full amount of time allotted (range: 5 minutes 7 seconds to 6 minutes). Only one group, in one of the two discussions, finished slightly earlier (4 minutes 27 seconds). All discussions were audio-recorded and subsequently transcribed. Recordings were deleted immediately after transcription to ensure anonymity and to allow for analysis of the discussions without any possibility of identifying participants.

In total, each participant had two individual and two group arguments. The questionnaire indicated whether the news was to be discussed individually or with other group members. Half of the participants would first have the two individual arguments and then the group arguments and the other half would start with the two group arguments. The four order of news presentation and the corresponding types of arguments are listed in full in the supplementary material (S1 File).

After each argument (regardless of the type), the participants had to answer some questions individually. First, they were asked to re-state their intention to share and their judgement about the truth of the news. They then answered some general questions about the experiment and the group experience. In particular, they were asked to indicate whether they knew the other participants with whom they had interacted during the group phases (categorized as a dummy variable where 0 indicates that the participant did not know anyone in the group, 1 indicates that the participant knew at least one person with whom they had interacted). Participants were also asked to rate the group experience by indicating on a scale ranging from 1 to 6 (where 1 meant "Absolutely not" and 6 meant "Absolutely yes") (a) whether they had been afraid that their own opinion would be evaluated negatively by the other members of the group discussion and (b) whether they had found the interaction pleasant. Finally, participants were asked to provide demographic information such as age, gender and education level.

## Results

First, we reversed the ratings for fake news so that higher scores consistently reflected greater accuracy for both real and fake news, with 1 representing no accuracy and 6 representing maximum accuracy.

As a preliminary check, we estimated a linear mixed-effects model with phase as a fixed effect and news item as a random intercept to account for item-specific variability. No significant main effect of phase emerged ($\beta = 0.11$, $SE = 0.088$, $F(1, 883) = 1.56$, $p = .212$), while item-level variance accounted for a notable proportion of the total variance (ICC = .125). To examine whether this variability reflected systematic differences between news types, we then included news type (fake vs real) and its interaction with phase as fixed effects in a second model. In this extended model, the phase × news type interaction was significant ($\beta = -0.491$, $SE = 0.176$, $F(1, 882) = 7.78$, $p = .005$), whereas the main effect of phase remained non-significant. The item-level variance was substantially reduced (ICC = .029), indicating that most of the between-item variability was associated with the distinction between fake and real news rather than idiosyncratic properties of specific items. Based on these results, we proceeded to calculate mean accuracy scores for fake and real news (see Fig 1) and conducted a 2 (news type: fake vs. real) × 2 (phase: 1 vs. 2) ANOVA to examine changes in accuracy across phases, with post hoc comparisons focusing on the dynamics of fake news.

The results showed a significant main effect of the news type ($F(1, 110) = 41.97$, $p < .001$, $\eta^2_p = .276$) but not of the phase ($F(1, 110) = 2.72$, $p = .102$, $\eta^2_p = .024$). The interaction between news type and phase was statistically significant ($F(1, 110) = 15.14$, $p < .001$, $\eta^2_p = .121$). Mean accuracy scores for fake news ranged from 3.86 in phase 1, 95% CI = [3.69, 4.04], to 4.22 in phase 2, 95% CI = [4.03, 4.41]; for real news, mean accuracy scores ranged from 3.30 in phase 1, 95% CI = [3.14, 3.46], to 3.15 in phase 2, 95% CI = [2.97, 3.33]. Post hoc comparisons using the Tukey HSD test showed that the improvement for fake news was statistically significant (mean difference = −0.36, $SE = .09$, $t = -4.08$, $p_{tukey} < .001$). By contrast, for real news, the difference between the mean scores in the two phases was not statistically significant (mean difference = 0.15, $SE = .09$, $t = 1.66$, $p_{tukey} = .350$).

Next, we examined whether the improvement in fake news accuracy scores was dependent on the type of argumentation (individually argued vs. discussed in groups). A repeated-measures ANOVA was conducted with phase and type of

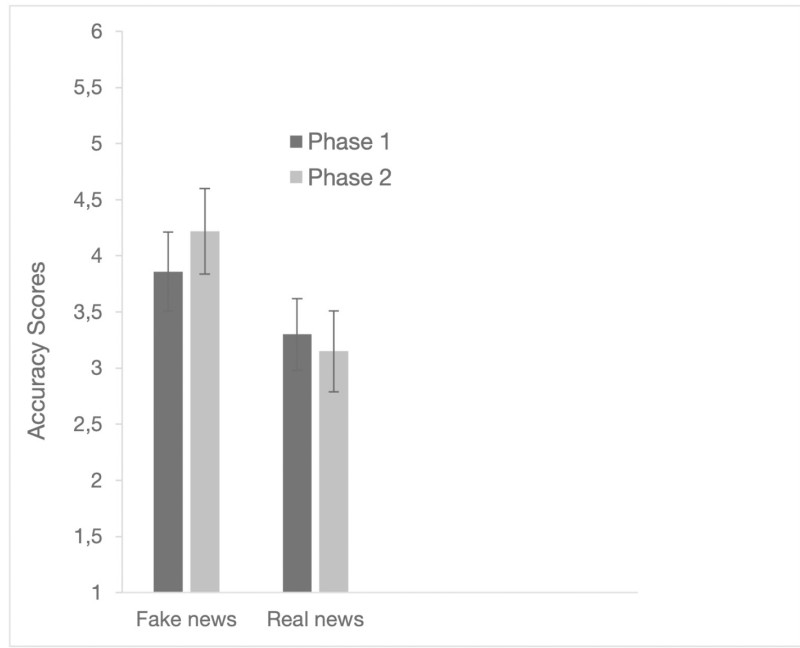

**Fig 1. Accuracy scores and 95% confidence intervals in phases 1 and 2 of fake news and real news.**

argumentation as within-subject factors. The results showed a significant main effect of phase ($F$(1, 110) = 16.68, $p<$.001, $\eta^2_p$ = .132) but no significant main effect of type of argumentation ($F$(1, 110) = 0.90, $p$ = .344, $\eta^2_p$ = .008). The interaction between phase and type of argumentation was statistically significant ($F$(1, 110) = 4.04, $p$ = .047, $\eta^2_p$ = .035). Mean accuracy scores for individually argued fake news ranged from 3.88 in phase 1, 95% CI = [3.65, 4.11], to 4.08 in phase 2, 95% CI = [3.85, 4.31]. In contrast, mean accuracy scores for fake news discussed in groups ranged from 3.84 in phase 1, 95% CI = [3.61, 4.06], to 4.35 in phase 2, 95% CI = [4.11, 4.59]. Post-hoc comparisons using the Tukey HSD test showed that the increase in accuracy was statistically significant only for fake news discussed in groups (mean difference = −0.51, $SE$ = .12, $t$ = −4.23, $p_{tukey}<$.001), whereas the change in accuracy for individually argued fake news was not significant (mean difference = −0.2, $SE$ = .11, $t$ = −1.75, $p_{tukey}$ = .300). These results are illustrated in Fig 2. Boxplots showing the distribution of accuracy scores (1–6 scale) across phases (Phase 1 vs. Phase 2) and argumentation type (Individual vs. Group) are also provided in the supplementary material (S1 Fig). As a robustness check, we also estimated a linear mixed-effects model including a random intercept for the groups the participants belonged to, applied to the first fake news item presented. This analysis supported the significant phase × argumentation type interaction (S2 File), indicating that the observed improvement in accuracy for group-discussed fake news was robust to group variability.

These results suggest that the increase in the ability to recognize fake news observed in phase 2 was specific to the group argumentation and was not merely a general effect of the opportunity for further reflection.

We then proceeded to test the existence of a possible transfer effect of group discussions on subsequent individual ability to recognize fake news. With respect to the type of argumentation, there were two orders: the so-called *G-order*, in which participants discussed the first two news stories as a group and then the last two individually, and the so-called *I-order*, in which the opposite was the case (the first two stories were discussed individually and the last two in groups). Two mixed ANOVAs – one for fake news discussed in groups and the other for fake news discussed individually – were conducted with phase as within-subject factor and the two different orders (*G* vs *I*) as between-subject factor. As expected, for individually argued fake news, the analysis revealed a significant interaction between phase and experimental order

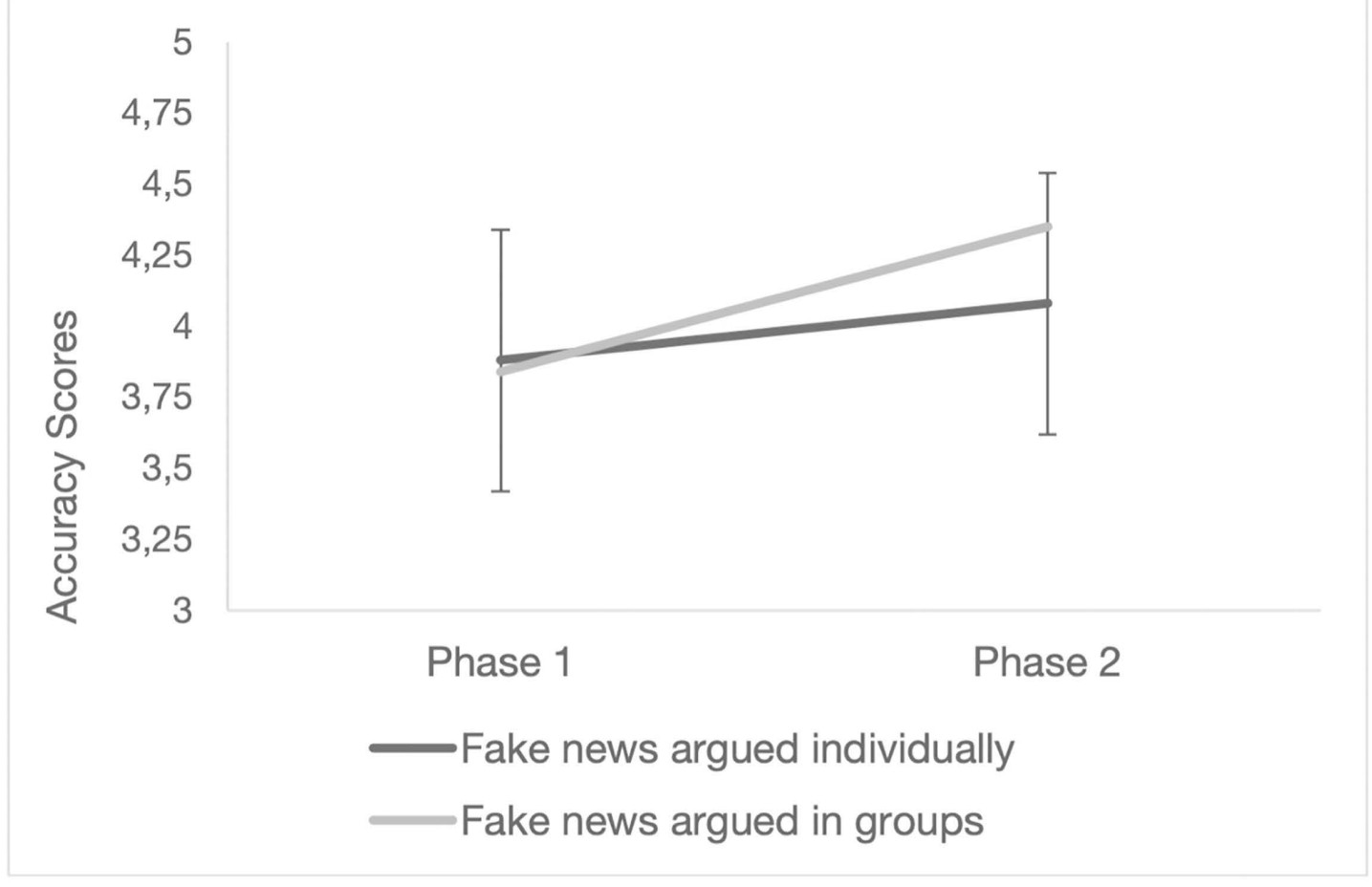

**Fig. 2. Accuracy scores and 95% confidence intervals in phases 1 and 2 of fake news discussed in groups and argued individually.**

($F$(1, 109) = 4.66, $p$ = .033, $\eta^2_p$ = .041). In *G-order*, mean accuracy for individually argued fake news went from 3.72 in phase 1, 95% CI = [3.39, 4.05], to 4.17 in phase 2, 95% CI = [3.84, 4.49]. In *I-order*, mean accuracy for individually argued fake news went from 4.04 in phase 1, 95% CI = [3.72, 4.35], to 4.00 in phase 2, 95% CI = [3.68, 4.32]. Post-hoc comparisons using the Tukey HSD test confirmed that the shift in *G-order* was significant (mean difference = −0.44, $SE$ = .16, $t$ = −2.79, $p_{tukey}$ = .031) whereas there was no significant difference between the two phases in *I-Order* (mean difference: 0.03, $SE$ = .15, $t$ = 0.23, $p_{tukey}$ = .996). For the fake news discussed in groups, the analysis showed no significant interaction between accuracy in the two phases and order ($F$(1, 109) = 0.87, $p$ = .354).

A significant interaction between phase and order observed only for individually argued fake news was consistent with the hypothesis of a transfer effect of the group discussion. It should be noted that, as the news items were presented in a partially counterbalanced design, each item was argued individually by half of the participants and discussed in a group by the other half. This enabled us to rule out the possibility that the observed effect was due to the content of a specific piece of fake news. Fig 3 shows these results. In the Supplementary material, separate graphs for Formaldehyde and Thyroid are also provided, along with the respective boxplots (S2 Fig, S3 Fig, S4 Fig, S5 Fig). The supplementary material also reports analyses for real news, which, as expected, did not show any order effect for the two types of argumentations on ability to recognize real news (S2 File).

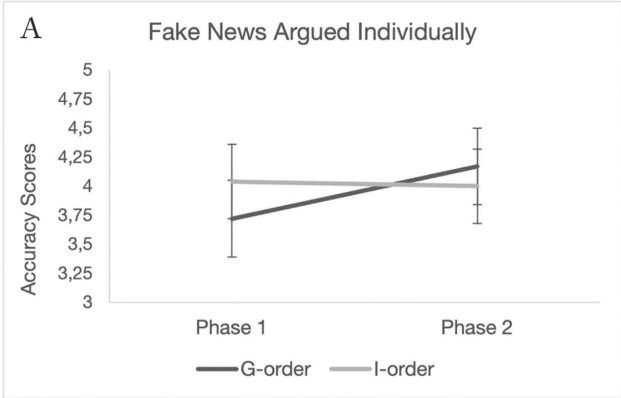
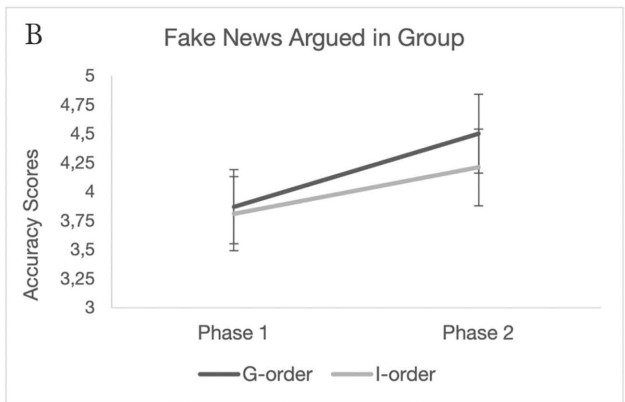

**Fig 3. Accuracy scores for fake news argued in group and fake news argued individually in G-order and in I-order. (A)** Individually argued fake news. **(B)** Group-discussed fake news. Error bars represent 95% confidence intervals.

One possible explanation for the effect observed in G-order was an increased engagement resulting from group discussions. To examine this possibility, using the number of words written in the individual arguments as a proxy for engagement, we first conducted a factorial nonparametric analysis using the aligned rank transform (ART) with news type (fake vs. real) as a within-subject factor and order (G vs. I) as a between-subject factor. The ART ANOVA revealed a significant main effect of order, $F(1, 108) = 17.28$, $p < .001$, which was qualified by a significant Order × News Type interaction, $F(1, 108) = 11.80$, $p < .001$; the main effect of news type was not significant, $F(1, 108) = 0.03$, $p = .856$. Based on this interaction, follow-up analyses were conducted separately for fake and real news using Kruskal–Wallis tests. For fake news, where the transfer effect was observed, the increase in the number of words ($M$ I-order = 57.9, $SD$ = 27.4; $M$ G-order = 71.9, $SD$ = 42.5) in individual arguments in G-order was not statistically significant, $\chi^2 (1) = 3.08$, $p = .079$, $\varepsilon^2 = .03$. In contrast, for real news—where neither a transfer effect on accuracy nor a general increase in accuracy at Phase 2 was detected—the increase in the number of words from I-order to G-order ($M$ I-order = 47.7, $SD$ = 22.2; $M$ G-order = 78.5, $SD$ = 39.0) was statistically significant $\chi^2 (1) = 23.20$, $p < .001$, $\varepsilon^2 = .21$. Taken together, these findings suggest that the G-order effect observed for fake news is unlikely to be attributable to increased engagement as indexed by the length of individual arguments.

To shed some light on the ways in which group argumentation affected fake news recognition, we then examined the role of the accuracy of the two people with whom participants discussed the news. We ran a hierarchical regression for group-argued fake news, with the accuracy score of the participant in phase 1 as predictor 1, the average score of the group partners in phase 1 as predictor 2, and the accuracy score of the participant in phase 2 as the dependent variable. The analyses showed that after controlling for individual accuracy, the mean accuracy of the two group partners in phase 1 explained a significantly greater proportion of the variance ($R^2$ predictor 1 = .212, $R^2$ predictor 2 = .293, $p < .001$; see Table 1). In other words, the quality of the two group partners – as measured by the accuracy ratings they gave when they first saw the messages (phase 1) – was a significant predictor of the accuracy ratings participants gave after the group discussion (phase 2).

We conducted the same analysis for individually argued fake news not preceded by group discussions (i.e., individually argued fake news in I-order). We ran a hierarchical regression for individually argued fake news, with the accuracy score of the participant in phase 1 as predictor 1, the mean score of the group partners in phase 1 as predictor 2, and the participant's accuracy score in phase 2 as the dependent variable. In this case, the inclusion of the mean accuracy of the two partners in phase 1 did not explain a significantly greater proportion of the variance ($R^2$ predictor 1 = .407, $R^2$ predictor 2 = .410, $p = .583$). Notably, for individually argued fake news, predictor 1 (participant's accuracy in phase 1) accounted for a higher proportion of variance ($R^2$ predictor 1 = .407) than for group-argued fake news ($R^2$ predictor 1 = .212). These

**Table 1. Hierarchical regression results for fake news discussed in group.**

| Variable | B | 95% CI for B | | SE B | β | R² | ΔR² |
|---|---|---|---|---|---|---|---|
| | | LL | UL | | | | |
| Step 1 | | | | | | .21 | .08*** |
| Constant | 2.49*** | 1.77 | 3.20 | 0.36 | | | |
| Participant's accuracy score in phase 1 | 0.49*** | 0.31 | 0.66 | 0.09 | .46*** | | |
| Step 2 | | | | | | .29 | .08*** |
| Group partners accuracy mean in phase 1 | 0.42*** | 0.18 | 0.65 | 0.12 | .29*** | | |

results suggest a greater tendency to confirm one's initial response when fake news items are argued individually. Specifically, an examination of the distributions of ratings retention versus change from phase 1 to phase 2 for the first fake news story showed that 52.6% of participants who argued the story individually retained their initial rating, compared to 29.6% of participants who discussed it in a group. This difference was statistically significant [$\chi^2(1, N=111) = 6.05$, $p=.014$].

But how the participants' ratings changed from phase 1 to phase 2 depending on the mean accuracy of the two group partners in phase 1? For the fake news discussing groups, we examined what occurred when the mean accuracy score of the two partners in phase 1 was higher than the participant's own accuracy score ($\Delta>0$), equal to it ($\Delta=0$), or lower ($\Delta<0$). As shown in Table 2, the large majority of participants (73% and 68%, respectively) made more accurate judgments in phase 1 (P2>P1) when the delta was greater than or equal to zero. However, when the delta was negative, only 36% of participants consistently changed their accuracy and made less accurate judgments in phase 1 (P2<P1). We coded the changes in participants' accuracy scores from phase 1 to phase 2 with respect to positive or negative delta values. Changes were coded as "consistent" if participants increased their accuracy when the delta was positive or decreased their accuracy when the delta was negative; "inconsistent" if participants decreased their accuracy when the delta was positive or increased their accuracy when the delta was negative; and "zero" when participants did not change their accuracy score. The $\chi^2$ test showed that the frequency of consistent changes was significantly higher when the average accuracy score of the two partners was higher than the participant's own accuracy score [$\chi^2(2, N=89) = 8.80$, $p=.012$].

Lastly, we examined group composition in terms of pre-discussion dissent. Based on members' phase 1 accuracy scores, groups were categorized as heterogeneous, F-homogeneous (i.e., all members in phase 1 already endorsed the falsity of the fake news subsequently discussed in the group), T-homogeneous (i.e., all members in phase 1 endorsed the truthfulness of the fake news subsequently discussed in group). In total, there were twenty-four heterogeneous groups, nine F-homogeneous groups and four T-homogeneous groups. We conducted a repeated-measures ANOVA with the accuracy scores for the fake news in phase 1 and phase 2 as the within-subjects factor, and group type as the between-subject factor. The analysis revealed no significant interaction between repeated measures (accuracy scores in the two phases) and group type ($F=1.01$; $p=.366$). From a purely descriptive perspective, in the four T-homogeneous groups in which all members initially endorsed the truthfulness of the fake news, 5 of the 12 participants (41.7%) judged

**Table 2. Frequencies and percentages of changes of accuracy scores from phase 1 to phase 2 as a function of the deltas between the accuracy of the group partners and the accuracy of the participant in phase 1.**

| Delta accuracy between participants * | P2<P1 | P2=P1 | P2>P1 | TOT |
|---|---|---|---|---|
| △>0 | 4 (7%) | 11 (20%) | 30 (73%) | 45 |
| △=0 | 1 (5%) | 6 (27%) | 15 (68%) | 22 |
| △<0 | 16 (36%) | 17 (39%) | 11 (25%) | 44 |

Note. *Delta between average accuracy score of the two partners and the participant's own accuracy score in phase 1

the item as false following group discussion. Thus, rather than exhibiting polarization toward the incorrect response, nearly half of participants of these groups shifted toward the correct answer.

## Intention to share

As for news sharing intention, we conducted a 2x2 ANOVA for phase (phase 1 and phase 2) and news type (real news and fake news). The results showed a significant main effect of the phase ($F(1, 110) = 5.81$, $p = .018$, $\eta^2_p = .05$) and of the news type ($F(1, 110) = 15.18$, $p < .001$, $\eta^2_p = .121$). In contrast, there was no statistically significant interaction between phase and news type ($F(1, 110) = 0.34$, $p = .560$, $\eta^2_p = .003$). The results are illustrated in the Figure 4.

   Overall, sharing levels were low, but a greater propensity to share real news emerged, with no difference between the two phases. Since the relationship between perceived accuracy and intention to share news is central to fake news research, we also examined the correlations between accuracy judgements and intention to share news in the two phases. For the two fake news items, intention-to-share and accuracy scores were strongly and negatively correlated in both phase 1 (Formaldehyde: $r(109) = -.49$, $p < .001$; Thyroid: $r(109) = -.49$, $p < .001$) and phase 2 (Formaldehyde: $r(109) = -.60$, $p < .001$; Thyroid: $r(109) = -.55$, $p < .001$). For the real news, the analyses for Masks showed a strong positive correlation between intention to share scores and accuracy in phase 1 ($r(109) = .41$, $p < .001$) and phase 2 ($r(109) = .63$, $p < .001$), while for Nutella there was no correlation between intention to share and accuracy in phase 1 ($r(109) = .12$, $p = .217$) and only a weak positive correlation in phase 1 ($r(109) = .21$, $p = .027$). These different results for the Nutella news item could be due to the peculiarity of the content of this news item, which is more likely to captivate participants and arouse their curiosity than the other news items.

## Other variables

Lastly, we checked whether the improvement in accuracy resulting from group argumentation could be influenced by social or emotional variables. We conducted a linear regression with the differences in accuracy scores between the two phases for fake news discussed in the group as the dependent variable, and the social and emotional variables (i.e., knowing others, fear of others' judgment, enjoyment of the interaction) as predictors. The results showed that none of the

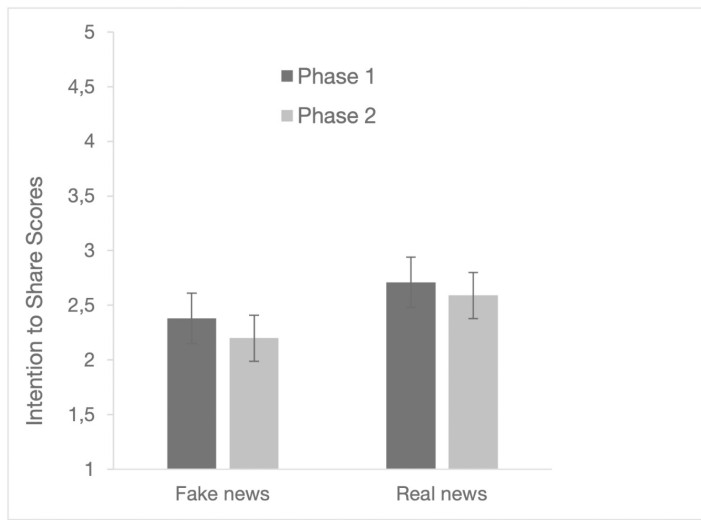

**Fig. 4. Intention to share and 95% confidence Intervals in phase 1 and 2 for fake news and real news.**

social or emotional variables significantly predicted the changes in accuracy scores between the two phases (all *ps* > .05). The full analyses can be found in the supplementary material (S2 Table).

## Discussion

We found evidence that participants' ability to recognize fake news improved after they had the opportunity to discuss the veracity of the information in small groups. Specifically, participants' judgements following a brief group discussion were more accurate than the judgements they made both prior to the discussion and when provided additional time to think individually. None of the social-emotional variables we assessed (i.e., familiarity with group members, fear of others' judgment, enjoyment of the interaction) were found to be predictive of these results. Therefore, the results appear to be consistent with the involvement of a social dimension of reasoning, highlighting the potential value of exchanging opinions, rather than being straightforwardly explained by the social interaction factors assessed here, such as mutual familiarity, fear of judgment, or enjoyment of the interaction.

Moreover, the positive effect of the group discussion seemed to extend to the subsequent individual assessments: when the first of the two fake news stories was discussed in a group, accuracy in judging the second fake news story (which was analyzed individually) increased significantly. Consistent with a transfer effect, this result suggests that the differences observed between judgments following individual or group-based reflection are unlikely to be attributable to differences in response bias. One potential explanation for this transfer effect could be increased motivation or engagement. The analysis of the number of words written in individual arguments—a variable we used as a proxy for engagement—however, revealed a dissociation between engagement and accuracy: participants wrote more words for real news than for fake news in G-order, yet the observed improvement in accuracy occurred exclusively for fake news. This pattern indicates that engagement, operationalized here as the length of written arguments, a measure commonly used as an index of cognitive or motivational investment in prior research [31,32], is unlikely to fully account for the observed transfer effect. Nevertheless, future research should incorporate mediating variables, such as argument quality or changes in cognitive strategies, to further examine the mechanisms underlying this effect.

Regarding the partners' contribution to post-discussion accuracy improvements, the average pre-discussion accuracy of the two other group members was a significant predictor. However, an asymmetric effect emerged: If the average pre-discussion accuracy score of the two partners was higher than the participant's own accuracy score, then 73% of the participants improved their accuracy after the discussion; conversely, when the average pre-discussion accuracy score of the two partners was lower than their own, only 36% of the participants exhibited a decrease in accuracy. These findings highlight the significant role of partner accuracy and pose challenges for alternative explanations that attribute the observed effects solely to more general social factors present in group contexts, such as conformity tendencies or social pressure.

Lastly, the group discussion led to an increase in accuracy only in the evaluation of fake news. This finding aligns with previous research [12] and is not surprising as even a weak argument may suffice to raise doubts about the veracity of a news item, while only factual evidence may convincingly prove its veracity. In sum, group discussions not only expose participants to others' pre-discussion assessments, but they also seem to highlight good arguments for recognizing fake news and provide insights that may be retained and subsequently drawn upon during individual assessment.

In general, our results are consistent with the tenets of argumentative theory and with findings from related research (e.g., [20],[33]), which demonstrate that argumentation plays a crucial role in the spread of correct responses, particularly when such responses are counterintuitive. For instance, Trouche and colleagues [20] showed that participants could identify the correct answer to a reasoning problem when it was proposed by a group member, discarding incorrect answers even if their own initial responses differed. They also found that participants could subsequently apply these correct arguments to similar problems, revealing a transfer effect similar to the pattern observed in our study. Moreover, among the factors influencing group discussions, their results showed that argumentative quality plays a more important role than participants' confidence. However, reasoning problems such as those used in these studies, which have a deductively

valid solution that is accessible to all participants – although often identified spontaneously by only a few – could provide optimal conditions for the benefits of argumentation and group discussion. In the present study, we observed similar results in the context of misinformation, where the validity of an answer cannot be definitively established and only arguments of varying quality can be put forward to support opposing viewpoints. Evidence of the group advantage in a judgement task came from Klein and Epley's [24] studies on lies detection, which found that real groups were more accurate than individuals, though the carry-over effect of pre-formed individual opinions limited the benefit in their study. Two factors could explain why Klein and Epley's participants were more likely to show a carry-over effect of their initial opinions than our participants. First, Klein and Epley's participants made binary judgments (i.e., whether a person was lying) that required a definite position, whereas our participants rated news accuracy on a six-point scale (most likely false /most likely true), which allowed for more nuanced responses. Second, although research has demonstrated the objective difficulty, people tend to overestimate their ability to detect lies [34].

Overall, these studies highlight the potential cognitive advantages of group discussion and the ability to transfer reasoning insights beyond the immediate group context, providing a broader framework for interpreting our results.

Nevertheless, our results might appear inconsistent with some evidence coming from fake news literature, such as Bago et al. [12], who found that individual deliberation reduces belief in false headlines. However, in that study, the perceived accuracy of the fake headlines after deliberation exceeded that of the initial intuitive judgments, but not the accuracy observed when participants judged the headlines without time constraints. In general, it is worth noting that our finding—that individual further reflection alone does not improve accuracy—is not inconsistent with recent literature on fake news, and in particular with the two key findings of the inattention-based account: the priming effect of accuracy on fake news sharing, and the relationship between individuals' propensity for analytical thinking and their ability to distinguish between true and false news [35]. It is likely that the initial judgements of our participants (phase 1) already reflected the best they could achieve through individual reflection. However, as the results of the present study demonstrate, accuracy can be further improved if participants engage in collaborative discussion with others.

The study has some limitations. First, it is a single study in which only four news items were used. Although the results exhibit internal consistency and align with the existing literature, and the balanced presentation of the items ensures that the reported effects are not attributable to specific content, questions about the generalizability of the findings remain, and future studies should expand the stimulus set.

Second, our sample consisted of university students, a population likely familiar with collaborative group work and potentially underrepresenting individuals most susceptible to misinformation. Additionally, the majority of participants were female (79%), which may limit the representativeness and generalizability of the findings. Future research should examine whether these results extend to more diverse populations and settings. Another potential limitation concerns the repeated exposure to the same news items in Phase 2, which raises the possibility of memory or familiarity effects. Because participants encountered each news item twice, improvements in accuracy could in principle reflect remembering earlier judgements or increased familiarity with the headlines. Prior research has shown that repeated exposure can sometimes increase belief in fake news through the illusory truth effect (e.g., [30,36,37]). However, this pattern is not consistent with our findings. In the present study, repeated exposure was associated with reduced belief in fake news, and this improvement occurred selectively following group discussion rather than uniformly across phases or items. If familiarity or mere exposure were driving the effects, one would typically expect either no change or an increase in perceived accuracy of fake news across conditions. Moreover, research indicates that familiarity effects are attenuated when individuals actively evaluate the accuracy of information at initial exposure [38] and that repetition does not reliably produce familiarity-driven backfire effects when corrective or evaluative processes are engaged [39]. These considerations suggest that memory or familiarity effects are unlikely to account for the observed improvements in accuracy, which instead appear to depend on the nature of the cognitive engagement elicited by group discussion and argumentation. A further limitation is the absence of pre-registration. As the study was exploratory in nature, building on existing literature while addressing a relatively

underexplored question, analytical choices were not specified in advance. While this approach is suitable for hypothesis generation, future confirmatory studies would benefit from pre-registered designs to enhance the robustness of the findings. Lastly, it should be noted that accuracy was assessed using a 1–6 Likert scale, which was treated as continuous in the analyses. While treating ordinal data as continuous is common and accepted in the literature [40–42], this approach may slightly affect the precision and interpretation of the results.

## Conclusions

Research on fake news has made considerable efforts to identify methods to improve individuals' ability to detect misinformation, but it has not yet investigated the extent to which individuals can support each other in this process. The present findings provide preliminary evidence in this direction, highlighting the potential benefits of structured group discussion within a tightly controlled experimental setting albeit with a limited set of stimuli. From a practical perspective, these results suggest that creating structured opportunities for discussion could represent an effective strategy to mitigate the pervasive influence of misinformation, particularly in shaping public attitudes towards science. Educational settings such as schools and universities could represent a promising starting point, as they offer collaborative environments in which argumentative skills can be developed. Such initiatives could provide students, and thus future generations, with valuable tools to critically engage with and counter misinformation in everyday life. Beyond these practical implications, the present findings also point to several directions for future research. First, future studies should replicate and extend the present findings by increasing the number of stimuli and involving more diverse populations. They could also examine the pattern observed in the present study that is consistent with a transfer effect, in order to assess its reproducibility across different materials. Finally, future research should investigate the mechanisms underlying the group argumentation effect by examining potential mediating variables such as argument quality, as well as possible social influences, for example by comparing written argumentation with active group discussion.

## Supporting information

**S1 File. Supplementary methods.** Full texts of the four news items and the four order of news presentation and the corresponding types of arguments.
(DOCX)

**S1 Table. Pre-test results on news items.**
(DOCX)

**S2 File. 1. Supplementary results.** Mixed-effects model for the first fake news item and analysis of the order for real news.
(DOCX)

**S2 Table. Linear regression output for emotional/social variables.**
(DOCX)

**S1 Fig. Distribution of accuracy scores for fake news by argumentation type and phase.**
(DOCX)

**S2 Fig. Accuracy scores for formaldehyde news item argued in group and formaldehyde news item argued individually and standard errors in G-order and in I-order.**
(JPG)

**S3 Fig. Accuracy scores for thyroid news item argued in group and thyroid news item argued individually and standard errors in G-order and in I-order.**
(JPG)

**S4 Fig. Distribution of accuracy scores for formaldehyde by argumentation type, phase and order.**
(DOCX)

**S5 Fig. Distribution of accuracy scores for thyroid by argumentation type, phase and order.**
(DOCX)

## Author contributions

**Conceptualization:** Diana Carbone, Donatella Ferrante.

**Data curation:** Diana Carbone.

**Formal analysis:** Diana Carbone, Francesco Marcatto.

**Investigation:** Diana Carbone, Donatella Ferrante.

**Methodology:** Diana Carbone, Donatella Ferrante.

**Project administration:** Donatella Ferrante.

**Software:** Francesco Marcatto.

**Supervision:** Donatella Ferrante.

**Writing – original draft:** Diana Carbone, Donatella Ferrante.

**Writing – review & editing:** Diana Carbone, Francesco Marcatto, Donatella Ferrante.

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
