## [Decision Letter · Decision Letter 0]

3 Dec 2025

PONE-D-25-43739Thinking Together: How Group Argumentation Boosts Fake News RecognitionPLOS ONE

Dear Dr. Carbone,

Thank you for submitting your manuscript to PLOS ONE. After careful consideration, we feel that it has merit but does not fully meet PLOS ONE’s publication criteria as it currently stands. Therefore, we invite you to submit a revised version of the manuscript that addresses the points raised during the review process.

We look forward to receiving your revised manuscript.

Kind regards,

Haixia Wang

Academic Editor

PLOS ONE

**Journal Requirements:**

2. Please note that your Data Availability Statement is currently missing the repository name and a direct link to access each database. If your manuscript is accepted for publication, you will be asked to provide these details on a very short timeline. We therefore suggest that you provide this information now, though we will not hold up the peer review process if you are unable.

**Additional Editor Comments:**

This study has certain theoretical and practical implications, and there are still some methodological details that require modification:

The definition of fake newsThe specific procedures of the experimentOther issues mentioned by the two reviewers

Reviewers' comments:

Reviewer's Responses to Questions

**Comments to the Author**

1. Is the manuscript technically sound, and do the data support the conclusions?

Reviewer #1: Yes

Reviewer #2: Partly

2. Has the statistical analysis been performed appropriately and rigorously?

Reviewer #1: Yes

Reviewer #2: No

3. Have the authors made all data underlying the findings in their manuscript fully available?

Reviewer #1: Yes

Reviewer #2: Yes

4. Is the manuscript presented in an intelligible fashion and written in standard English?

Reviewer #1: Yes

Reviewer #2: Yes

5. Review Comments to the Author

Reviewer #1: The paper appears to adopt a novel approach; however, the author struggles to clearly define the concept of fake news. A precise definition has not yet been provided. Additionally, the author uses the term misinformation inconsistently, without establishing its contextual meaning, and shifts abruptly to disinformation, which carries a distinct implication. Therefore, it is suggested that the author refer to substantial prior works on the subject to develop clear conceptual distinctions among fake news, misinformation, and disinformation, and accordingly revisit and refine the literature review and data analysis, if necessary. They may refer/revisit works of

1. Wardle, C. Fake News (2017) It’s Complicated, First Draft News, https://firstdraftnews.com/fake-news-complicated/

2. European Commission Contribution to the European Council (2018) Action Plan Against

Disinformation. Brussels, 5 December, JOIN (2018) 36 final, https://ec.europa.eu/commission/

sites/beta-political/files/eu-communication-disinformation-euco-05122018_en.pdf/

3. UK, House of Commons, Digital, Culture, Media, and Sport Committee (2019) Disinformation

‘Fake News’: Final Report. https://publications.parliament.uk/pa/cm201719/cmselect/cmcumeds/1791/1791.pdf/

4. Commisso, C. (2017) ‘The post-truth archive: considerations for archiving context in fake news

repositories’, Preservation, Digital Technology and Culture, Vol. 46, No. 3, pp.99–102,

doi: https://doi.org/10.1515/pdtc-2017-0010.

Reviewer #2: This manuscript reports a within-subjects experiment (N = 111) examining whether group discussion enhances the ability to recognize fake news compared to individual written reflection. Participants judged the accuracy of two real and two fake news items in Phase 1, then re-evaluated the same items in Phase 2 after either a group discussion or an individual argumentation exercise. The key finding is that accuracy improved only for fake news discussed in groups, not for items argued individually. The authors also report a transfer effect: discussing one fake news item in a group increased accuracy for subsequently judged (but individually evaluated) fake news items.

The study is well motivated, conceptually interesting, and contributes to the literature on misinformation and the social dimension of reasoning. It addresses a theoretically meaningful question: Do group discussions actually make people better at detecting misinformation? The general pattern of results is clear and has implications for both theory and practice.

However, several issues limit the manuscript’s clarity, robustness, and generalizability. I outline these below.

Major Comments

1. Limited number of stimuli

The study uses only four news items (two real, two fake). This severely restricts claim generalizability and inflates the influence of item idiosyncrasies (e.g., the “Nutella” real-news anomaly). Although the authors acknowledge this limitation, the problem is more serious than stated.

Recommendations:

• Treat results as preliminary; tone down conclusions accordingly.

• Include item-level analyses (e.g., mixed-effects models with random intercepts for items) if possible, even with small item numbers, to demonstrate the robustness of effects.

• Consider adding these data to a future preregistered replication with a substantially larger item pool.

2. Interpretation of “analytical thinking” is not directly supported

The manuscript repeatedly claims that group argumentation “stimulates analytical thinking” and that improvements occur because reasoning processes are activated. However, no measures of analytical thinking (CRT, need for cognition, verbal reasoning markers, etc.) are included.

Thus, this explanation is speculative.

Recommendations:

• Reframe claims to avoid strong causal statements about analytical thinking.

• Alternatively, incorporate linguistic analysis of written arguments (e.g., argument quality, evidence use) to support the interpretation.

• At minimum, clearly distinguish results from interpretation: the data show group-improved accuracy; the theoretical lens is Argumentative Theory.

3. Methodological clarity in the group-discussion procedure

The instructions suggest that:

• Participants presented arguments one at a time, in experimenter-determined order,

• Followed by free discussion.

But several details are unclear:

• Did all groups actually use the full 6 minutes?

• Were experimenters present and visible?

• Were discussions recorded and coded? If not, why not?

• How were interruptions, dominance, or social loafing controlled for?

Because group dynamics are central to the claim, more detail is needed.

Recommendations:

• Provide more complete procedural information.

• Consider reporting descriptive features of groups (average talking time, whether all participants contributed, etc.), even qualitatively.

4. Statistical approach could be improved

a. ANOVA may be suboptimal

Given the within-subjects structure and non-independent items, a mixed-effects model (participants × items) would be more appropriate.

b. Accuracy scores are ordinal

A 1–6 scale is treated as continuous. This is common but should be acknowledged as a limitation.

c. Multiple comparisons

Several post-hoc tests are conducted. Corrections (Tukey is used in places, but inconsistently reported) should be applied consistently and transparently.

5. Order effects deserve deeper exploration

The “transfer effect” is one of the most interesting findings, but the interpretation is limited.

Questions needing clarification:

• Did the transfer effect occur for both fake items or only one?

• Could this be an artifact of stimulus difficulty order rather than group discussion?

• Did participants in the group-first condition also show more engagement or longer response times, suggesting increased task motivation rather than reasoning transfer?

Suggestions:

• Provide item-specific plots for each order condition.

• Analyze whether Phase 2 response times differ by order and argumentation type.

• Discuss alternative explanations more fully.

6. Potential confounds in Phase 2

Participants saw the same news items again. This raises at least two concerns:

a. Memory effects

Accuracy improvements could reflect remembering earlier judgments.

b. Familiarity effects

Prior exposure may reduce belief in real news (a known finding) and can increase belief in fake news (illusory truth effect). The study acknowledges neither.

While the authors argue that individual reflection does not cause improvement, the design still conflates repeated exposure with deliberation.

Recommendation:

• Include a discussion of familiarity and exposure effects as alternative mechanisms.

7. Social factors may be under-examined

The authors report that social variables (fear of judgment, familiarity with group members, enjoyment of discussion) did not predict gains. But:

• The items used to measure these constructs are single-item measures with unknown reliability.

• The study does not examine whether certain group compositions (e.g., mixed-accuracy triads) produce more benefit.

Given the strong role of group quality in predicting improvement, more attention to social processes seems warranted.

Minor Comments

1. Pre-registration

There is no mention of preregistration. Because the analytic choices (e.g., averaging across items) are somewhat flexible, preregistration would strengthen the contribution.

2. Figures could be clearer

o CI bars overlap in many places, making visual interpretation difficult.

o Consider adding raw distribution plots or violin/boxplots.

3. Terminology:

The term “phase” is accurate but could be more explicitly defined earlier in the manuscript.

4. Stimuli selection:

The rationale for choosing specific real/fake items could be more thoroughly justified beyond simple pretesting.

5. Generalisability:

The sample (mostly young Italian students, 79% female) limits applicability to other populations; this should be highlighted.

Overall Evaluation

This is an interesting paper with a clear result: group argumentation improves fake news detection more than individual reflection does. The study aligns with findings from Argumentative Theory and meaningfully extends the literature by applying group-reasoning paradigms to misinformation.

However, the manuscript requires:

• clearer methodological detail,

• more cautious interpretation,

• a more thorough exploration of alternative explanations,

• improved statistical framing, and

• stronger discussion of limitations.

With revisions, the manuscript has potential to contribute meaningfully to the field.

6. PLOS authors have the option to publish the peer review history of their article (what does this mean?). If published, this will include your full peer review and any attached files.

Reviewer #1: No

Reviewer #2: No

---

## [Author Response · Author response to Decision Letter 1]

17 Jan 2026

EDITOR

Haixia Wang, Ph.D.

Academic Editor

PLOS ONE

Thank you for submitting your manuscript to PLOS ONE. After careful consideration, we feel that it has merit but does not fully meet PLOS ONE’s publication criteria as it currently stands. Therefore, we invite you to submit a revised version of the manuscript that addresses the points raised during the review process.

Dear Editor, we thank you and the reviewers for the time and effort devoted to the evaluation of our manuscript entitled “Thinking Together: How Group Argumentation Boosts Fake News Recognition” (Manuscript ID: PONE-D-25-43739). We are pleased to submit a revised version of the manuscript, in which we have carefully addressed all the comments and suggestions provided during the review process.

As required, this response letter details how each comment has been addressed and how the manuscript has been revised accordingly. Our responses are provided in blue font below each comment.

Additional Editor Comments: This study has certain theoretical and practical implications, and there are still some methodological details that require modification:

1. The definition of fake news

2. The specific procedures of the experiment

3. Other issues mentioned by the two reviewers

We thank the Editor for highlighting the need for greater conceptual clarity and methodological detail. In response, we have revised the manuscript to clarify the definition and use of the terms fake news and misinformation, and to ensure terminological consistency throughout the paper. In addition, we have expanded the description of the experimental procedures and addressed the methodological and interpretative issues raised by the reviewers. All changes are described in detail below and highlighted in the revised manuscript.

Reviewer 1

The paper appears to adopt a novel approach; however, the author struggles to clearly define the concept of fake news. A precise definition has not yet been provided. Additionally, the author uses the term misinformation inconsistently, without establishing its contextual meaning, and shifts abruptly to disinformation, which carries a distinct implication. Therefore, it is suggested that the author refer to substantial prior works on the subject to develop clear conceptual distinctions among fake news, misinformation, and disinformation, and accordingly revisit and refine the literature review and data analysis, if necessary. They may refer/revisit works of

1. Wardle, C. Fake News (2017) It’s Complicated, First Draft News, https://firstdraftnews.com/fake-news-complicated/

2. European Commission Contribution to the European Council (2018) Action Plan Against

Disinformation. Brussels, 5 December, JOIN (2018) 36 final, https://ec.europa.eu/commission/

sites/beta-political/files/eu-communication-disinformation-euco-05122018_en.pdf/

3. UK, House of Commons, Digital, Culture, Media, and Sport Committee (2019) Disinformation

‘Fake News’: Final Report. https://publications.parliament.uk/pa/cm201719/cmselect/cmcumeds/1791/1791.pdf/

4. Commisso, C. (2017) ‘The post-truth archive: considerations for archiving context in fake news

repositories’, Preservation, Digital Technology and Culture, Vol. 46, No. 3, pp.99–102,

doi: https://doi.org/10.1515/pdtc-2017-0010.

We thank the reviewer for highlighting the need to provide a clear definition of fake news and for emphasizing the importance of using related terms consistently throughout the manuscript. In response, we have added a dedicated paragraph in the Introduction clarifying how these terms are defined and used in the manuscript. We explicitly draw on prior influential work in the field (European Commission, 2018; UK & House of Commons Digital, Culture, Media, and Sport Committee, 2018; Wardle, 2017) as well as a recent systematic review providing a unified taxonomical framework (Kapantai et al., 2021). In particular, we now specify that we use the term fake news as a generic descriptor for news-like content containing false information, without reference to the intentionality of its source. Misinformation is used to refer in general to false or inaccurate information shared without deliberate intention to deceive. Terminology has been revised consistently throughout the manuscript, including the keywords and headings. References to disinformation have been removed, as the study does not address intentionality and the term was therefore not appropriate in this context.

Importantly, because the study does not assess or manipulate the intentionality behind the false information used as stimuli, these conceptual distinctions—while theoretically relevant—do not directly bear on the operationalization of the stimuli or on the analytical strategy adopted. We therefore believe that the main analyses and conclusions remain robust with respect to these distinctions. These revisions can be found in the Introduction (p. 3) and throughout the manuscript where terminology has been harmonized.

Reviewer 2

This manuscript reports a within-subjects experiment (N = 111) examining whether group discussion enhances the ability to recognize fake news compared to individual written reflection. Participants judged the accuracy of two real and two fake news items in Phase 1, then re-evaluated the same items in Phase 2 after either a group discussion or an individual argumentation exercise. The key finding is that accuracy improved only for fake news discussed in groups, not for items argued individually. The authors also report a transfer effect: discussing one fake news item in a group increased accuracy for subsequently judged (but individually evaluated) fake news items.

The study is well motivated, conceptually interesting, and contributes to the literature on misinformation and the social dimension of reasoning. It addresses a theoretically meaningful question: Do group discussions actually make people better at detecting misinformation? The general pattern of results is clear and has implications for both theory and practice.

However, several issues limit the manuscript’s clarity, robustness, and generalizability. I outline these below.

We thank the reviewer for the careful and thorough evaluation of our manuscript. We greatly appreciate the detailed feedback, which has been extremely helpful in improving the clarity, rigor, and overall quality of the paper.

We have carefully considered each comment and observation and have made corresponding changes in the revised version of the manuscript. In the sections below, we provide a detailed, point-by-point explanation of how each observation has been addressed.

Major Comments

1. Limited number of stimuli

The study uses only four news items (two real, two fake). This severely restricts claim generalizability and inflates the influence of item idiosyncrasies (e.g., the “Nutella” real-news anomaly). Although the authors acknowledge this limitation, the problem is more serious than stated.

Recommendations:

• Treat results as preliminary; tone down conclusions accordingly.

• Include item-level analyses (e.g., mixed-effects models with random intercepts for items) if possible, even with small item numbers, to demonstrate the robustness of effects.

• Consider adding these data to a future preregistered replication with a substantially larger item pool.

We thank the reviewer for these important observations regarding the limited number of news items used in the study. We agree that the small number of stimuli (two real and two fake news items) imposes constraints on the generalizability of the findings and may increase the influence of item-specific idiosyncrasies.

In response, we have taken several steps to address this concern:

1. Cautious framing of conclusions.

We have revised the manuscript to adopt a more cautious tone. In particular, we have revised the Discussion by grounding our interpretations more explicitly in the relevant literature and by expanding the limitations section. We have also reviewed the wording of several statements throughout the manuscript. For example, the sentence in the Conclusions previously stating that the study “provides the first evidence in this direction” has been replaced with “provides preliminary evidence in this direction,” thereby more accurately reflecting the exploratory nature of the study.

2. Item-level analyses.

To address the potential impact of item-related variability on our results, we removed the initial analysis that was intended to control for content effects (a 2 × 2 ANOVA by news content and phase, previously reported in the Supplementary Materials). We instead conducted linear mixed-effects models with phase as a fixed effect and news item as a random intercept to account for item-specific variability. No significant main effect of phase emerged, and item-level variance accounted for a notable proportion of the total variance (ICC = .125). When news type (fake vs. real) and its interaction with phase were included as fixed effects, the phase × news type interaction was significant, and item-level variance was substantially reduced (ICC = .029), indicating that most between-item variability was associated with news type rather than idiosyncratic properties of individual items (p. 10).

3. Future preregistered replication.

We have also added a note in the discussion acknowledging that the study was exploratory and did not include preregistration. We suggest that future studies could replicate these findings with a larger pool of news items in a preregistered design to enhance robustness.

We believe these revisions address the reviewer’s concerns, clarify the exploratory scope of the study, and demonstrate that the observed effects are robust with respect to the idiosyncratic properties of individual items.

2. Interpretation of “analytical thinking” is not directly supported

The manuscript repeatedly claims that group argumentation “stimulates analytical thinking” and that improvements occur because reasoning processes are activated. However, no measures of analytical thinking (CRT, need for cognition, verbal reasoning markers, etc.) are included.

Thus, this explanation is speculative.

Recommendations:

• Reframe claims to avoid strong causal statements about analytical thinking.

• Alternatively, incorporate linguistic analysis of written arguments (e.g., argument quality, evidence use) to support the interpretation.

• At minimum, clearly distinguish results from interpretation: the data show group-improved accuracy; the theoretical lens is Argumentative Theory

We thank the reviewer for this important comment. By using the term analytical thinking, we intended to refer only to the idea that group discussion may foster a more critical evaluation of news content. However, we agree that this use of the term could generate ambiguity. Therefore, we have revised the manuscript to remove explicit references to analytical thinking throughout the text, including the Abstract and headings. The interpretation of the findings has been reframed within the framework of Argumentative Theory, which is used as a theoretical perspective to interpret the results rather than as a claim about specific underlying cognitive processes.

The revised manuscript more clearly distinguishes between the empirical findings—showing improved accuracy following group discussion—and their theoretical interpretation, thereby better aligning the scope of the conclusions with the available data.

3. Methodological clarity in the group-discussion procedure

The instructions suggest that:

• Participants presented arguments one at a time, in experimenter-determined order,

• Followed by free discussion.

But several details are unclear:

• Did all groups actually use the full 6 minutes?

• Were experimenters present and visible?

• Were discussions recorded and coded? If not, why not?

• How were interruptions, dominance, or social loafing controlled for?

Because group dynamics are central to the claim, more detail is needed.

Recommendations:

• Provide more complete procedural information.

• Consider reporting descriptive features of groups (average talking time, whether all participants contributed, etc.), even qualitatively.

We thank the reviewer for highlighting the need for more detailed information regarding the group-discussion procedure. In response, we have revised the Methods section to provide a clearer and more complete description of the procedure.

Specifically, we now clarify that:

• The group discussion began with a structured turn-taking phase, in which participants presented their arguments one after the other in the experimenter-determined order. This ensured that every participant had the opportunity to express their view, reducing potential effects of dominance, interruptions, or social loafing.

• The experimenter was visible only during the initial instruction phase, kept the camera off during the discussion, and intervened only when strictly necessary (e.g., in the event of prolonged silence).

• All members contributed at least once during the initial turn-taking phase.

• Almost all groups used the full 6 minutes allotted (range: 5 minutes 7 seconds to 6 minutes), with only one exception (4 minutes 27 seconds in a single discussion).

• All discussions were audio-recorded and subsequently transcribed in anonymized form. Recordings were deleted immediately after transcription.

These revisions provide more complete procedural transparency and address the reviewer’s concerns regarding the handling of group dynamics and the quality of participation.

4. Statistical approach could be improved

a. ANOVA may be suboptimal

Given the within-subjects structure and non-independent items, a mixed-effects model (participants × items) would be more appropriate.

b. Accuracy scores are ordinal

A 1–6 scale is treated as continuous. This is common but should be acknowledged as a limitation.

c. Multiple comparisons

Several post-hoc tests are conducted. Corrections (Tukey is used in places, but inconsistently reported) should be applied consistently and transparently.

We thank the reviewer for these thoughtful methodological comments, which helped us improve the clarity and robustness of the statistical analyses. We address each point in turn.

(a) Use of mixed-effects models.

We agree that mixed-effects models are appropriate given the within-subjects design and the non-independence introduced by group discussion. We addressed this concern in two complementary ways.

First, as reported in the Results section (p. 10), we conducted preliminary linear mixed-effects models including news item as a random intercept to assess the extent to which item-specific idiosyncrasies influenced the observed effects. These analyses showed that although item-level variance was present in a baseline model, it was largely explained by the distinction between fake and real news once news type and its interaction with phase were included. This supported our decision to aggregate accuracy scores by news type in the main analyses, while explicitly acknowledging the limited number of stimuli as a limitation in the Discussion.

Second, we conducted an additional robustness analysis for the effect of the type of argumentation using a linear mixed-effects model with random intercepts for group, reported in the Supplementary Materials (S2 File). Group was specified as a random effect to explicitly control for the possibility that the effect of argumentation type depended on systematic differences between groups (e.g., some groups being generally more accurate than others). By modeling group-level variance, we assessed whether belonging to a particular discussion group accounted for a substantial portion of the variance beyond the fixed effect of argumentation type. Although the intraclass correlation coefficient indicated non-negligible between-group variability, the phase × argumentation type interaction remained statistically significant, suggesting that the improvement observed for fake news discussed in groups cannot be attributed solely to group-specific characteristics.

This mixed-effects analysis was restricted to the first fake news item presented, in line with the study’s experimental aims. Because the study explicitly tested whether group discussion could

---

## [Decision Letter · Decision Letter 1]

26 Feb 2026

PONE-D-25-43739R1Thinking Together: How Group Argumentation Boosts Fake News RecognitionPLOS One

Dear Dr. Carbone,

Thank you for submitting your manuscript to PLOS ONE. After careful consideration, we feel that it has merit but does not fully meet PLOS ONE’s publication criteria as it currently stands. Therefore, we invite you to submit a revised version of the manuscript that addresses the points raised during the review process.

The paper closely engages with the theoretically and practically significant question of whether group argumentation aids in the identification of fake news, addressing a gap in current misinformation research concerning the social dimension of cognition. However, the following issues still require consideration:

(1) Sample representativeness remains limited.

The sample consists primarily of Italian university students, with a disproportionately high percentage of female participants (79%), which may affect the generalizability of the findings. It is recommended that this limitation be more explicitly acknowledged in the discussion, and that future research validate the findings across more diverse populations.

(2) The number of stimulus items is still limited (four news items).

Although the authors used mixed-effects models to control for content heterogeneity, it remains difficult to fully rule out the influence of item-specific characteristics. It is recommended that the conclusions continue to emphasize the preliminary nature of the findings, and that future studies expand the stimulus set.

(3) The mechanism underlying the "transfer effect" requires further investigation.

The study found that individuals became more accurate in judging another fake news item after engaging in group discussion—a highly interesting transfer effect. However, its underlying mechanism remains unclear (e.g., transfer of cognitive strategies vs. increased motivation). It is recommended that this effect be interpreted more cautiously in the discussion, and that future research incorporate mediating variables (such as argument quality or changes in cognitive strategies) to further examine the mechanisms at play.

We look forward to receiving your revised manuscript.

Kind regards,

Haixia Wang

Academic Editor

PLOS One

Journal Requirements:

Reviewers' comments:

Reviewer's Responses to Questions

**Comments to the Author**

1. If the authors have adequately addressed your comments raised in a previous round of review and you feel that this manuscript is now acceptable for publication, you may indicate that here to bypass the “Comments to the Author” section, enter your conflict of interest statement in the “Confidential to Editor” section, and submit your "Accept" recommendation.

Reviewer #1: All comments have been addressed

Reviewer #2: (No Response)

2. Is the manuscript technically sound, and do the data support the conclusions?

Reviewer #1: Yes

Reviewer #2: Yes

3. Has the statistical analysis been performed appropriately and rigorously?

Reviewer #1: I Don't Know

Reviewer #2: Yes

4. Have the authors made all data underlying the findings in their manuscript fully available?

Reviewer #1: Yes

Reviewer #2: Yes

5. Is the manuscript presented in an intelligible fashion and written in standard English?

Reviewer #1: Yes

Reviewer #2: Yes

6. Review Comments to the Author

Reviewer #1: Following the discussion section, a conclusion may be included highlighting the key contributions and major findings of the study, along with suggestions and directions for future research.

Reviewer #2: The authors have responded thoroughly and in good faith to the previous round of major revisions. The manuscript is substantially improved, and the additional analyses, procedural clarifications, and expanded discussion largely address my earlier concerns. I thank the authors for the care and seriousness with which they engaged with the review.

Before final acceptance, I recommend a small number of minor, editorial revisions, focused on interpretive clarity rather than additional analyses:

Generalisability and robustness.

Although the manuscript now appropriately acknowledges the limited number of stimuli and adopts a more cautious tone overall, there are still a few places where the language implies a level of generality or robustness that exceeds what can reasonably be inferred from a four-item design. I recommend ensuring that claims, particularly in the Abstract, Conclusions, and discussion of mixed-effects results, are consistently framed as preliminary and specific to the present paradigm.

Framing of the transfer effect.

The additional analyses and item-level plots strengthen the case that the observed order-related improvement is not driven by a single stimulus. However, the transfer effect remains interpretive rather than definitive. I suggest maintaining consistent caution in how this effect is described (e.g., “consistent with” or “suggestive of” transfer) across the Results and Discussion.

Scope clarification.

A brief, explicit sentence clarifying the scope of the findings, for example, that they demonstrate the potential benefits of structured group discussion within a tightly controlled, small-stimulus experimental context, would further help align conclusions with the evidential base.

These are minor wording and framing adjustments only and do not require new data or analyses. Subject to these revisions, I support acceptance of the manuscript.

7. PLOS authors have the option to publish the peer review history of their article (what does this mean?). If published, this will include your full peer review and any attached files.

Reviewer #1: No

Reviewer #2: **Yes:** Piers Howe

---

## [Author Response · Author response to Decision Letter 2]

24 Mar 2026

EDITOR

Haixia Wang, Ph.D.

Academic Editor

PLOS ONE

Dear Editor,

We would like to thank you and the reviewers for the time and effort devoted to the evaluation of our manuscript entitled “Thinking Together: How Group Argumentation Boosts Fake News Recognition” (Manuscript ID: PONE-D-25-43739R1). We are pleased to submit a revised version of the manuscript, in which we have addressed the additional comments and suggestions provided during the latest round of review.

As requested, this response letter details how each comment has been addressed and how the manuscript has been revised accordingly. Our responses are provided in blue font below each comment, and all modifications in the manuscript are highlighted using track changes.

Editor Comments

The paper closely engages with the theoretically and practically significant question of whether group argumentation aids in the identification of fake news, addressing a gap in current misinformation research concerning the social dimension of cognition. However, the following issues still require consideration:

(1) Sample representativeness remains limited.

The sample consists primarily of Italian university students, with a disproportionately high percentage of female participants (79%), which may affect the generalizability of the findings. It is recommended that this limitation be more explicitly acknowledged in the discussion, and that future research validate the findings across more diverse populations.

We appreciate the editor’s comment regarding sample representativeness. The limitations section now explicitly acknowledges the limited representativeness of the sample. We clarified that the sample consisted of university students and that the majority of participants were female (79%), which may limit the representativeness and generalizability of the findings. We also noted that future research should examine whether the results extend to more diverse populations and settings.

(2) The number of stimulus items is still limited (four news items).

Although the authors used mixed-effects models to control for content heterogeneity, it remains difficult to fully rule out the influence of item-specific characteristics. It is recommended that the conclusions continue to emphasize the preliminary nature of the findings, and that future studies expand the stimulus set.

We agree with the editor that the limited number of stimuli should be explicitly acknowledged. The limitations section now notes that the study relied on a small number of stimuli (four news items). We also clarified that future studies should expand the stimulus set in order to further examine the robustness and generalizability of the findings. In addition, the concluding paragraph emphasizes the preliminary nature of the results within a tightly controlled experimental setting, albeit with a limited set of stimuli.

(3) The mechanism underlying the "transfer effect" requires further investigation.

The study found that individuals became more accurate in judging another fake news item after engaging in group discussion—a highly interesting transfer effect. However, its underlying mechanism remains unclear (e.g., transfer of cognitive strategies vs. increased motivation). It is recommended that this effect be interpreted more cautiously in the discussion, and that future research incorporate mediating variables (such as argument quality or changes in cognitive strategies) to further examine the mechanisms at play.

We thank the editor for this comment. In the discussion section we now explicitly acknowledge that the mechanism underlying the transfer effect remains unclear and that further research is needed to examine it. We have therefore added a sentence suggesting that future research should incorporate mediating variables (e.g., argument quality or changes in cognitive strategies) to further investigate the mechanisms underlying this effect.

Reviewer 1

Following the discussion section, a conclusion may be included highlighting the key contributions and major findings of the study, along with suggestions and directions for future research.

We appreciate the reviewer’s suggestion. The manuscript now includes a concluding paragraph summarizing the main findings and highlighting the implications of structured group discussion for improving misinformation detection. We also outline directions for future research, including the need to examine whether the results generalize to more diverse populations and contexts, to further investigate the pattern observed in the present study that is consistent with a transfer effect, and to explore potential mediating variables (such as argument quality) as well as possible social influences (for example, by comparing exposure to written arguments with active participation in group discussions).

Reviewer 2

The authors have responded thoroughly and in good faith to the previous round of major revisions. The manuscript is substantially improved, and the additional analyses, procedural clarifications, and expanded discussion largely address my earlier concerns. I thank the authors for the care and seriousness with which they engaged with the review.

Before final acceptance, I recommend a small number of minor, editorial revisions, focused on interpretive clarity rather than additional analyses:

We are very pleased with the reviewer’s positive assessment and would like to thank them once again for their thoughtful comments, which have helped us improve the manuscript substantially. We respond to the additional comments below.

1. Generalisability and robustness.

Although the manuscript now appropriately acknowledges the limited number of stimuli and adopts a more cautious tone overall, there are still a few places where the language implies a level of generality or robustness that exceeds what can reasonably be inferred from a four-item design. I recommend ensuring that claims, particularly in the Abstract, Conclusions, and discussion of mixed-effects results, are consistently framed as preliminary and specific to the present paradigm.

2. Framing of the transfer effect.

The additional analyses and item-level plots strengthen the case that the observed order-related improvement is not driven by a single stimulus. However, the transfer effect remains interpretive rather than definitive. I suggest maintaining consistent caution in how this effect is described (e.g., “consistent with” or “suggestive of” transfer) across the Results and Discussion.

3. Scope clarification.

A brief, explicit sentence clarifying the scope of the findings, for example, that they demonstrate the potential benefits of structured group discussion within a tightly controlled, small-stimulus experimental context, would further help align conclusions with the evidential base.

We thank the reviewer for these helpful suggestions. Following the reviewer’s recommendation, we revised the wording throughout the manuscript to ensure that the findings are consistently framed in a cautious manner and interpreted as preliminary within the context of a small-stimulus experimental design. In particular, we softened several formulations in the Abstract, Results, and Discussion (e.g., replacing “evidence that group discussion improves” with “preliminary evidence that group discussion may improve”, and describing the observed pattern as “consistent with a transfer effect” rather than implying a definitive transfer mechanism). We also revised the concluding paragraph to clarify the scope of the findings and to emphasize that they derive from a tightly controlled experimental setting albeit with a limited set of stimuli.

We hope that the revisions adequately address the concerns raised during the review process, and we would like to thank the editor and the reviewers again for their constructive feedback.

---

## [Editor Report · Decision Letter 2]

15 Apr 2026

Thinking Together: How Group Argumentation Boosts Fake News Recognition

PONE-D-25-43739R2

Dear Dr. Carbone,

We’re pleased to inform you that your manuscript has been judged scientifically suitable for publication and will be formally accepted for publication once it meets all outstanding technical requirements.

Kind regards,

Haixia Wang

Academic Editor

PLOS One
---

## [Editor Report · Acceptance letter]

PONE-D-25-43739R2

PLOS One

Dear Dr. Carbone,

I'm pleased to inform you that your manuscript has been deemed suitable for publication in PLOS One. Congratulations! Your manuscript is now being handed over to our production team.

Kind regards,

on behalf of

Dr. Haixia Wang

Academic Editor

PLOS One